# Research on the Dynamic Characteristics of Mechanical Seal under Different Extrusion Fault Degrees

Yin Luo *, Yakun Fan, Yuejiang Han, Weqi Zhang and Emmanuel Acheaw

Research Center of Fluid Machinery Engineering and Technology, Jiangsu University, Zhenjiang 212013, China; 2221811033@stmail.ujs.edu.cn (Y.F.); 2221711011@stmail.ujs.edu.cn (Y.H.); 2221811018@stmail.ujs.edu.cn (W.Z.); 5103190336@stmail.ujs.edu.cn (E.A.)

* Correspondence: luoyin@ujs.edu.cn; Tel.: +86-132-2262-6939

**Abstract:** In order to explore the dynamic characteristics of the mechanical seal under different fault degrees, this paper selected the upstream pumping mechanical seal as the object of study. The research established the rotating ring-fluid film-stationary ring 3D model, which was built to analyze the fault mechanism. To study extrusion fault mechanism and characteristics, different dynamic parameters were used in the analysis process. Theoretical analysis, numerical simulation, and comparison were conducted to study the relationship between the fault degree and dynamic characteristics. It is the first time to research the dynamic characteristics of mechanical seals in the specific extrusion fault. This paper proved feasibility and effectiveness of the new analysis method. The fluid film thickness and dynamic characteristics could reflect the degree of the extrusion fault. Results show that the fluid film pressure fluctuation tends to be more intensive under the serious extrusion fault condition. The extrusion fault is more likely to occur when the fluid film thickness is too large or too small. Results illustrate the opening force is affected with the fluid film lubrication status and seal extrusion fault degrees. The fluid film stiffness would not always increase with the rotating speed growth. The seal fault would occur with the increasing of rotating speeds, and the leakage growth fluctuations could reflect the fault degree.

**Keywords:** mechanical seal; dynamic characteristics; extrusion fault; numerical simulation; sealing performance; fluent

## 1. Introduction

As a widely used fluid transmission device, centrifugal pump plays a crucial role in the national economy. The proportion of pumps equipped with mechanical seals in industry has risen to 80%. The ratio is even greater among petrochemical industry, affecting up to 90%. Besides, statistics reveal that the fault caused by mechanical seals accounts for more than 40% in all machine faults. According to the statistics of centrifugal pump faults from the German Engineering Association [1], the sealing fault ratio is 31%, the rolling bearing fault ratio is 20%, the leakage fault ratio is 10%, the motor fault ratio is 10%, the rotor fault ratio is 9%, and the sliding bearing fault ratio is 8%. The sealing fault can cause an unpredictable waste of resources [2–4]. Mechanical seal faults can affect the internal flow of the centrifugal pump [5–7]. A serious situation would cause casualties and property losses. Thus, the research on mechanical seal fault diagnosis is necessary to ensure reliability and safety [8–10]. Moreover, it is necessary to research the dynamic characteristics detection of each stage when the mechanical seal fault occurs.

Great efforts have been made to do research on the dynamic characteristics of mechanical seals [11–14]. He et al. [15] studied that the viscous shear heat and frictional heat due to asperity

contact decrease with an increase in the thickness of the tapered film. As the shaft decelerates, the wear distance rate increases with an increase in the axial stiffness. The axial damping only affects the duration of the oscillations. Zhang et al. [16] researched that lubrication reduces friction and wear and generates heat, but leakage has to be considered. The effects of the sealing surface characteristics on the leakage, and the effects of the external factors of the sealing device on the leak rate. Towsyfyan et al. [17] mainly introduces the fault detection of mechanical seal friction by acoustic emission technology. Zhang et al. [18] investigated the fluidic leak rate through metal sealing surfaces by developing fractal models for the contact process and leakage process. Gropper et al. [19] provides a comparative summary of different modeling techniques for fluid flow, cavitation, and micro-hydrodynamic effects. Migout et al. [20] studied the relationship between the temperature change of the sealing medium and the vaporization of the medium under the seal rings face deformation condition, and pointed out that the stability of the fluid film would be seriously affected when the temperature change gradient of the medium is large. Varney et al. [21] researched the influence of the installation misalignment of the seal rings, established a three-degree-freedom dynamic model of the stationary ring, analyzed the response characteristics formed by the force excitation in all directions, and pointed out that the increase of the excitation intensity would lead to the occurrence of the collision phenomenon of the seal rings. Zhu et al. [22] studied that the sealing performance is enhanced by increasing the spacing of adjacent sealing sheets. The sealing sheets with positive bending angle have less air resistance in the flow path, which would lead to larger leakage. The increase in the number of sealing sheets gives rise to an increase in the generation probability of recirculation zone and vortex, which aggravates the mainstream energy loss. Mosavat et al. [23] researched that the influences of the thermal radiation on the temperature distribution of the mechanical face seal are investigated. Also, the effect of the stretching and shrinking on the thermal performance of the fin with different profiles are comprehensively studied.

Three-dimensional models of single-cone and double-cone were established [24]. A comparison made between the flow velocity, a shear rate and shear stress in single-cone and double-cone zones. This paper revealed the CFD analysis of the flow of a polymeric material inside the double-cone plasticization-homogenization zone of the screw-disc extruder.

Plenty of investigation and discussion have been undertaken on the development of sealing performance, temperature change, and distribution of mechanical seal. At present, there is a lack of monitoring research on mechanical seal failure state. The difference of this paper is to judge the fault degree of mechanical seal according to the dynamic characteristics.

Current research about the dynamic characteristics of mechanical seals mainly focuses on the seal rings modality variation and internal flow field analysis of mechanical seals. However, there is deficiency existing in mechanical seal fault though pressure, opening force, and the leakage. Therefore, the dynamic characteristics and fault mechanism of mechanical seal in different stages need to be studied urgently.

## 2. Analysis of Sealing Fault Mechanism

### 2.1. Normal State

As shown in Figure 1, mechanical seal is an important part to prevent leakage in centrifugal pumps. Mechanical seal model in normal state is shown in Figure 1. The rotating and stationary rings stick to each other. The function principle of mechanical seal is that a thin fluid film is formed between the rotating ring and the stationary ring. Hydrodynamic effects would be formed because of the thin fluid film. The fluid film with proper thickness can improve the lubrication performance and seal performance. A certain distance between the rotating ring and the stationary ring would be formed because of the hydrodynamic effects. The normal state is that the thickness between rotating ring and stationary ring is proper. At this time, the fluid film could provide enough open force to prevent the direct contact between the rotating ring and stationary ring. The fluid film would form a certain

resistance to prevent the medium from leaking out, and make the seal surface lubricated, so as to achieve the better sealing effect.

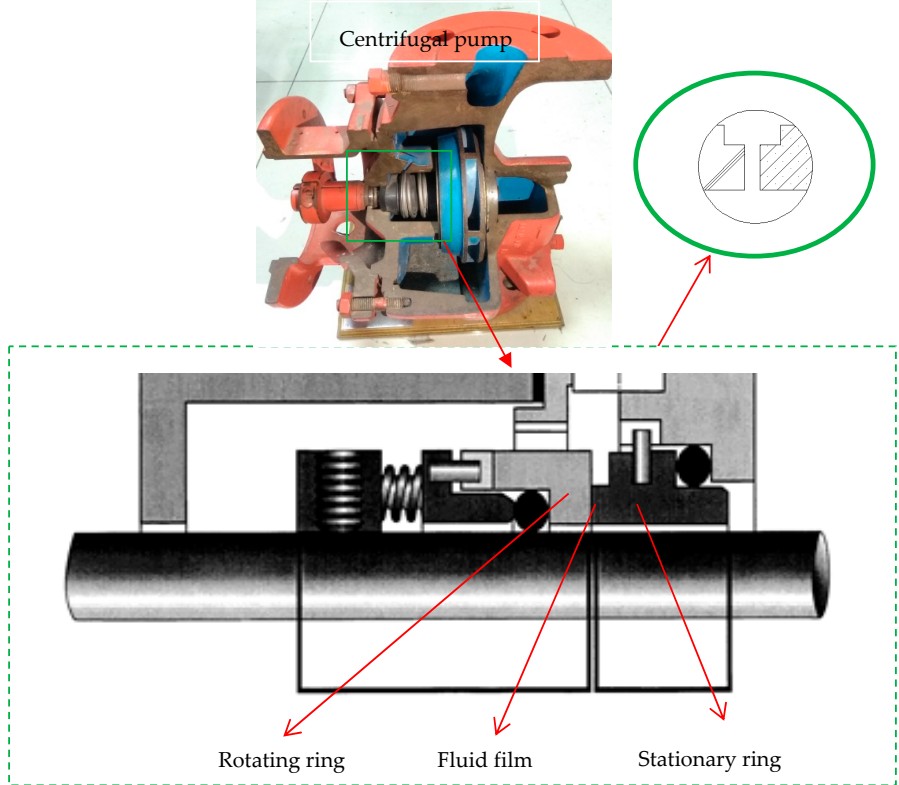

**Figure 1.** Schemes of mechanical seal model.

## 2.2. Fault State

The mechanical seal would often be in fault due to that the friction and wear behavior are universal phenomena in the rotating parts. Seal fault could be divided into many kinds, including seal surface extrusion fault, face temperature resistance fault, lateral load increasing. Many faults are caused by the squeezing of rotating and stationary rings. Figure 2 depicts the fault schematic diagrams, which show the extrusion fault in different degrees. The cause of the extrusion fault on the mechanical seal face might result from the relative deviation of the two seal ring surfaces during operation, the misaligned installation of the mechanical seal, or the mechanical seal external pressure.

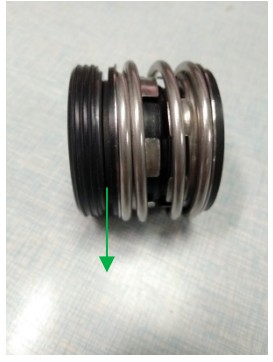
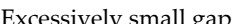
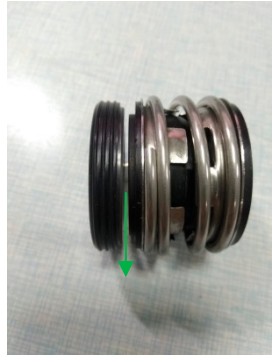
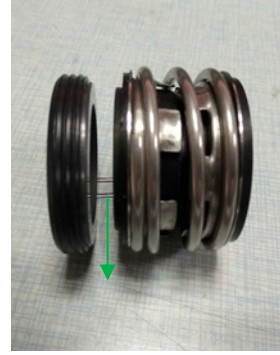

Excessively small gap     Normal state     Excessive clearance

**Figure 2.** Physical pictures of different fault degrees.

## 2.3. Extrusion Faults

This paper mainly studied the fault caused by the rotating and stationary ring extrusion of the mechanical seal. The mechanical seal would form a stable fluid film during normal operation. However, the stable fluid film would be destroyed, when the mechanical seal is in fault state. From normal to fault operation state, the internal hydrodynamic pressure and the force of seal rings would change. When the relative movement of the rotating and stationary rings results in extrusion fault, the slight faults would affect the sealing performance and the serious faults would result in extrusion deformation, wear damage, and fault of the rotating and stationary rings.

Figure 3 showed that the characteristics and change of the rotating ring surface under different fault degrees. Figure 3a indicated that the seal surface would be smooth and intact in normal state. Abrasion and damage would be found in the rotating ring surface when the seal is in slight faults, which is showed in Figure 3b. The crack caused by light extrusion could be seen in the red circle of Figure 3c. Significant damage and obvious behaviors would occur when the seal was in severe faults, which are illustrated in Figure 3c. The end face damage caused by severe extrusion fault could be seen in the red circle in Figure 3c.

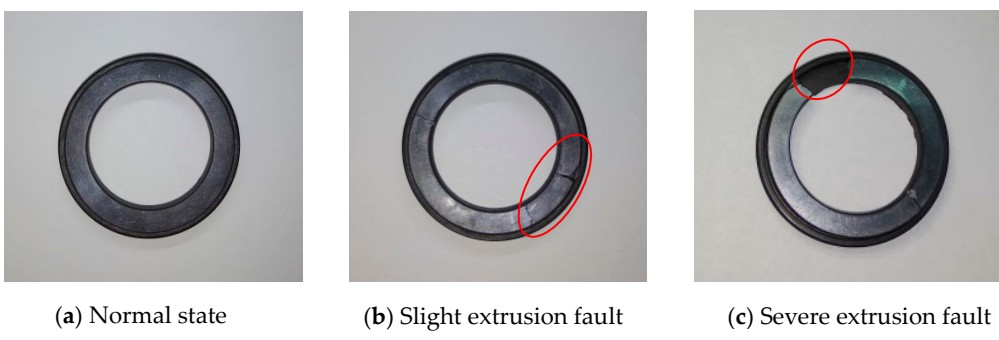

(**a**) Normal state　　　　　(**b**) Slight extrusion fault　　　　　(**c**) Severe extrusion fault

**Figure 3.** Rotating ring in different extrusion fault.

Change in mechanical seal rings was too complicated to be expressed by mathematical equations. Fault physical models were necessary to be carried out to describe the corresponding relationship between the extrusion fault and mechanical seal characteristics.

## 3. Establishment of Calculation Model

### 3.1. Fault Physical Models

In order to study the causes of mechanical seal fault and the internal sealing mechanism, it was necessary to establish a reasonable method and a proper fault physical model for analysis. The actual mechanical seal fault problem was affected by various factors. Therefore, the physical model of mechanical seal fault should be simplified. The fault of mechanical seal was mostly caused by extrusion wear of rotating and stationary rings, which resulted from the long-term uneven stress and the change of relative position of rotating and stationary rings during the long-term operation. In this paper, a simplified physical model of mechanical seal fault was established to analyze the extrusion fault of rotating and stationary rings. To establish the fault physical model, two aspects need to be considered. One is to assume that the relative movement of the rotating and stationary rings only occurs along the axial direction, and the movement along the radial direction was assumed to be zero. The second is to assume that the material texture of the rotating and stationary ring is uniform. In this paper, the thickness of the fluid film between the rotating and stationary rings was used to represent the distance between the rotating and stationary rings when different extrusion faults occur in the axial direction of the mechanical seal. The dynamic characteristics of fluid film with different thickness could reflect the different fault degrees of mechanical seal when extrusion fault occurs. Finally, the fault model of mechanical seal under this fault was formed.

In this paper, the upstream pumping mechanical seal was selected as the research object. The physical model was composed of three main parts: rotating ring, fluid film, and stationary ring. Figure 4 showed the axial section diagram of the rotating ring modeling and the geometric parameters of the section. The main geometrical parameters of the fluid film were selected: the inner radius of the rotating rings $r_i$ = 25 mm, the outer radius of the rotating ring $r_o$ = 32 mm. Figure 5 indicated the axial section diagram of the stationary ring modeling and the geometric parameters of the section. The inner radius of the stationary ring $r_i$ = 24.5 mm, the outer radius of the rotating ring $r_o$ = 31 mm. Figure 6 showed the three-dimensional diagram of fluid film modeling. Besides, the dimensions of the inner and outer rings of the fluid film were shown in detail. Because the order of magnitude in the direction of film thickness was micrometer, Figure 6 showed the enlarged fluid film thickness. The thickness of the fluid film was set to 1, 2, 3, 4, 5, 6, 7, 8, 9 μm, respectively. Figure 7 illustrated the computational domains of the mechanical seals.

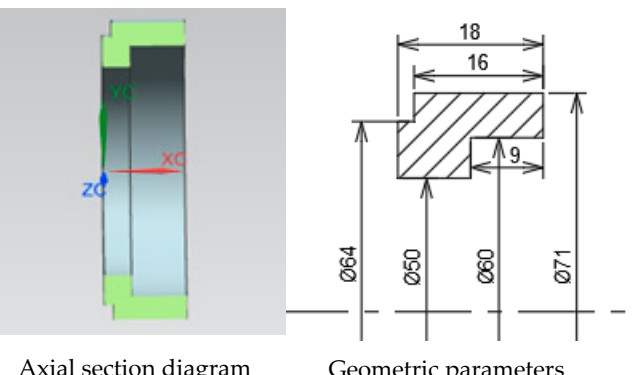

Axial section diagram   Geometric parameters

**Figure 4.** Rotating ring modeling diagram.

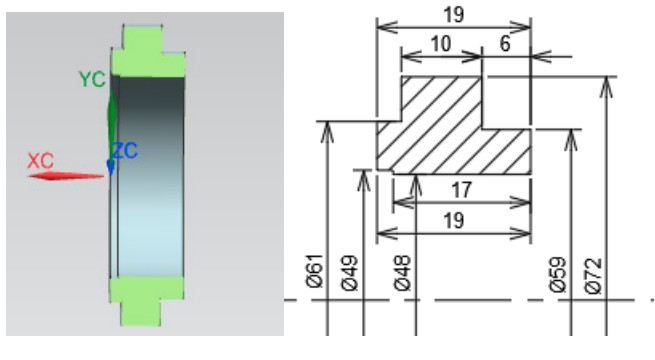

Axial section diagram   Geometric parameters

**Figure 5.** Stationary ring modeling diagram.

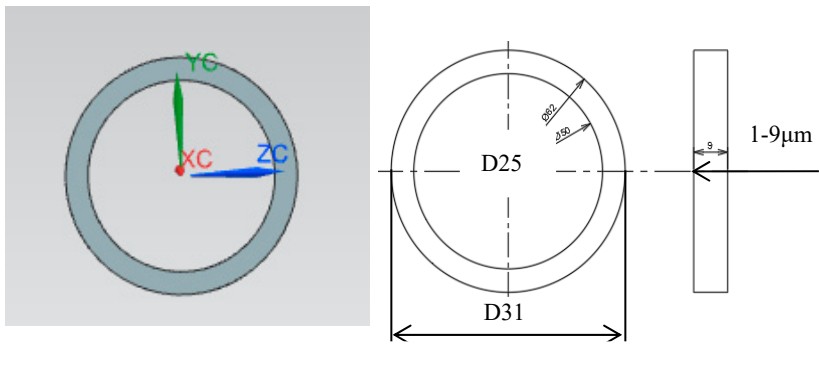

Radial section diagram   Geometric parameters

**Figure 6.** 3D diagrams of fluid film modeling.

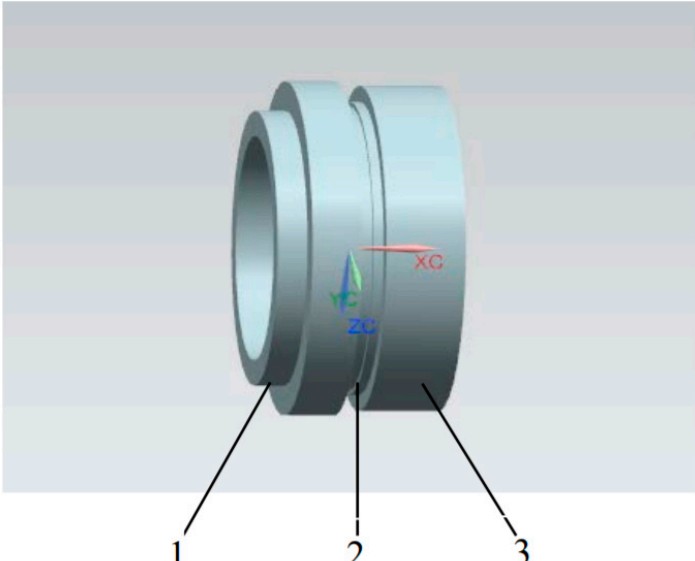

**Figure 7.** Computational domains of the mechanical seal (1-stationary ring, 2-fluid film, 3-rotating ring).

This section may be divided by subheadings. It should provide a concise and precise description of the experimental results, their interpretation as well as the experimental conclusions that can be drawn.

### 3.2. Dynamic Calculation Model

The fault of the mechanical seal was mostly manifested in the squeeze and wear fault of the rotating and stationary rings. The dynamic calculation model was selected according to the simplified fault physical model given above when the rotating and stationary rings fail due to extrusion fault. With the deepening of theoretical research on mechanical seals, the mechanical seal faults include various sciences such as mechanics, power, fluids, materials, chemistry, heat transfer, and tribology. In order to study the law between the mechanism and structural characteristics of mechanical seal when extrusion fault occurs, a dynamic calculation model suitable for fault physical model was adopted in this paper.

In order to make the simulation simple and clear, the following assumptions were made in this paper, considering the existence of fluid pressure, elastic force, solid deformation and the interaction force and heat transfer between fluid and solid in the mechanical seal.

- Mechanical seal consists of rotating ring, fluid film, stationary ring, and auxiliary system, the rotating ring part was provided with elastic force by the elastic element. The fluid film was formed by the liquid between the rotating and stationary ring. The stationary ring is a static part of the mechanical seal. The mechanical seal is simplified to simulate the process of faults conveniently. Thus, it is simplified into three parts: rotating ring, stationary ring, and fluid film.
- Because the distance between the rotating ring and the stationary ring is quite close and the thickness of the fluid film is tiny in the actual work of mechanical seal, the fluid film conforms to the Newton's law of viscosity and the effect of volume force and inertia force is ignored.
- The sealing medium is generally an incompressible fluid, but the density would change with pressure in this paper.
- It is assumed that the heat generated by friction is only transferred between the rotating and stationary rings of the mechanical seal, and the heat loss caused by stirring, thermal radiation, and leakage is ignored.
- It is assumed that the fluid has no velocity slip on the solid boundary.

### 3.2.1. Control Equations

The flow and diffusion of the liquid film fluid inside the mechanical seal satisfies the momentum equation, energy equation, and continuity equation [25].

1.  Fluid Domain Equation

The mass conservation equation

$$\frac{\partial \rho_f}{\partial t} + \frac{\partial}{\partial x_j}(\rho_f v_i) = 0 \tag{1}$$

In the above formula: $\rho_f$ is the fluid density; $v$ is the fluid velocity, subscript $i$, $j$ = 1, 2, 3, representing the components in three directions, $t$ is the time.

The momentum equation

$$\frac{\partial(\rho_f v_i)}{\partial t} + \frac{\partial}{\partial x_j}(\rho_f v_i v_j) = -\frac{\partial p}{\partial x_i} + \frac{\partial}{\partial x_j}(\mu \bullet \frac{\partial v_i}{\partial x_j}) \tag{2}$$

In the above formula: $p$ is the pressure of fluid film, $\mu$ is the dynamic viscosity of the fluid.

The energy equation

$$\frac{\partial \rho E}{\partial t} + \nabla \bullet [v(\rho E + p)] = \nabla \bullet \left[ k_{eff} \nabla T - \sum_j h_j J_j + (\Gamma_{eff} \bullet v) \right] + S_h \tag{3}$$

In the above formula: E is the total energy of the fluid micelle, $\rho$ is the density of the fluid micelle, and $p$ is the pressure of fluid film. $\Gamma_{eff}$ is the effective stress of fluid domain, $h_j$ is the enthalpy of the component, $K_{eff}$ is the effective thermal conductivity of fluid film, $J_j$ is the diffusion flux of the component $j$, $S_h$ is the source term including other volumetric heat.

2.  Solid Domain Equation

$$M_s \ddot{d} + C_s \dot{d} + K_s d + \tau_s = 0 \tag{4}$$

In the equation, $M_s$ represents for the mass matrix of the solid element, $C_s$ represents for the damping matrix of the solid element, $K_s$ represents for the rigidity matrix of the solid element, $d$ represents for the displacement vector of the solid element, $\tau_s$ represents for the stress on the rotating ring and stationary ring.

At the same time, the thermal deformation term caused by the temperature difference in the solid area as followed.

$$f_T = \alpha_T \bullet \nabla T \tag{5}$$

In the formula: $\alpha_T$ is the coefficient of thermal expansion related to temperature. $T$ is the temperature of the seal rings.

3.  Dynamic Model of Axial Movement

Based on the analysis of the kinematic relationship, the dynamical model of mechanical seal system was derived by using the D'Alembert principle. The force and axial movement of the mechanical seal are shown in Figure 8.

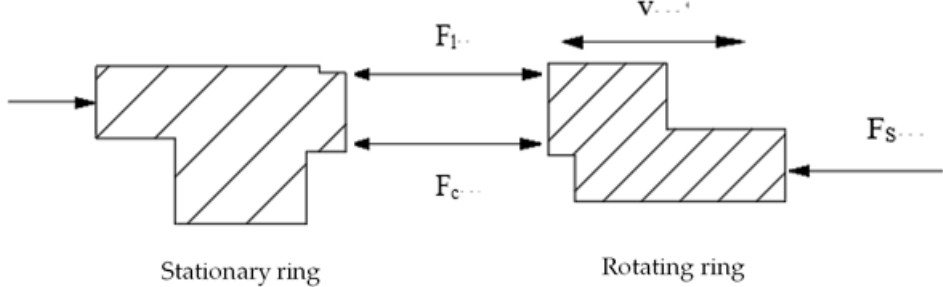

**Figure 8.** Axial dynamic model of mechanical seal.

From the dynamic point of view, the dynamic equation of mechanical seal axial movement was established as follows [26].

$$m\ddot{Z}_s + c\dot{Z}_s + F_s = F_l + F_c \tag{6}$$

In the formula: $m$ is the equivalent mass of the rotating ring and the spring. $Z_s$ is the axial relative displacement between the rotating ring and the stationary ring. $c$ is the axial damping coefficient of spring and auxiliary seal rings. $F_s$ is the force of the spring on the rotating ring. $F_l$ is the axial force of the fluid film between the end faces of the rotating ring and stationary ring, and $F_c$ is the axial contact force between the rotating ring and stationary ring.

### 3.2.2. Sealing Performance Parameters

1. Seal Opening Force

The opening force of mechanical seal surface is the sum of the pressure exerted on the seal surface by the liquid film fluid. The opening force could be obtained by integrating the pressure field of the liquid film on the seal surface.

$$F_0^\Omega = \int\int p dA = \int\int p r dr d\theta \tag{7}$$

In the formula: $p$ is the pressure of fluid film, $r$ is the radial coordinate, and $\Omega$ is the whole calculation area.

2. Leakage of Mechanical Seal

Leakage is an important indicator for measuring the performance of mechanical seal. Leakage $Q$ could be synthesized by this formula [27].

$$Q = \frac{\pi d_m h_o^3 \Delta p}{12\mu b} \tag{8}$$

$Q$-leakage, $d_m$-average diameter of the sealing surface, $h_o$-the thickness of fluid film, $\Delta_p$-pressure difference, $\mu$-dynamic viscosity of the medium, $b$-effective width of the seal.

### 4. Dynamic Simulations and Analysis

In this paper, the computational domains were meshed using the structured blocking hexahedral method. Due to the large difference between the radial and axial length of the fluid film parameters, the diameter of the fluid film is millimeter, but the thickness of the fluid film is micron, thus it is difficult to directly divide the three-dimensional mesh of the liquid film. Therefore, the fluid film corresponding to the center angle of 8.5 degrees was first meshed in this paper, and then the entire fluid film is formed based on this array. Figure 9 revealed coupling relationship among multiple fields in the dynamic simulations. To ensure the computational accuracy, mesh independence analysis was

conducted. Calculation of fluid film parameters with different number of grids was shown in the Table 1. The velocity was not growing when the number of grids increased to 3.6 million. As shown in Figure 10, the pressure became relatively stable after increasing the grid number to 3.6 million elements. After the grid independence test, the final grid number unit was 362,943. The fluid membrane grids were shown in Figure 11. In order to remove the influence of the order of magnitude of radial and axial fluid film grids on the simulation results, a grid-independent assessment was conducted. The numerical results were obtained by dividing the fluid film into different mesh numbers and repeating the simulation. The error of the final results is less than 1%, which shows that the mesh division has no effect on the simulation results.

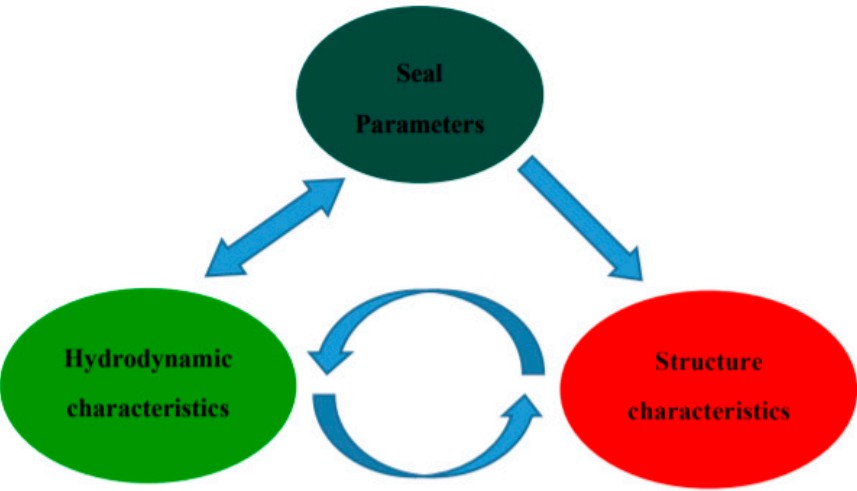

**Figure 9.** Multi-field coupling.

**Table 1.** Calculation of fluid film parameters with different number of grids.

| Grid Number ($\times 10^5$) | Pressure (Mpa) | Velocity (m/s) |
| :---: | :---: | :---: |
| 3.43 | 0.1615 | 6.74 |
| 3.51 | 0.1782 | 6.81 |
| 3.62 | 0.1825 | 6.85 |
| 3.67 | 0.1834 | 6.85 |
| 3.74 | 0.1852 | 6.85 |
| 3.85 | 0.1873 | 6.85 |
| 3.87 | 0.1869 | 6.85 |

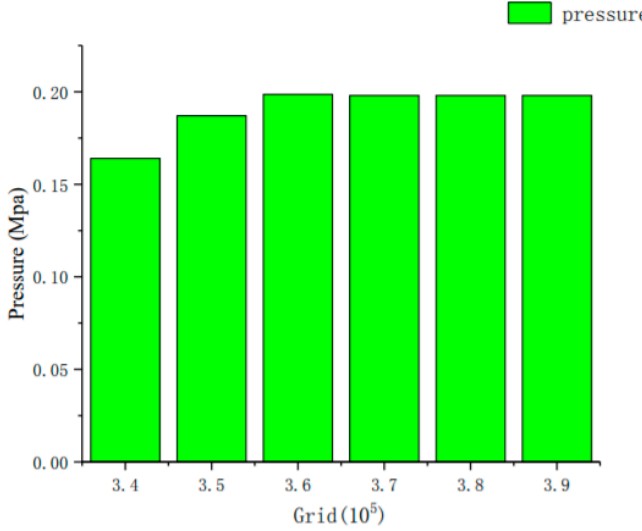

**Figure 10.** Mesh independence.

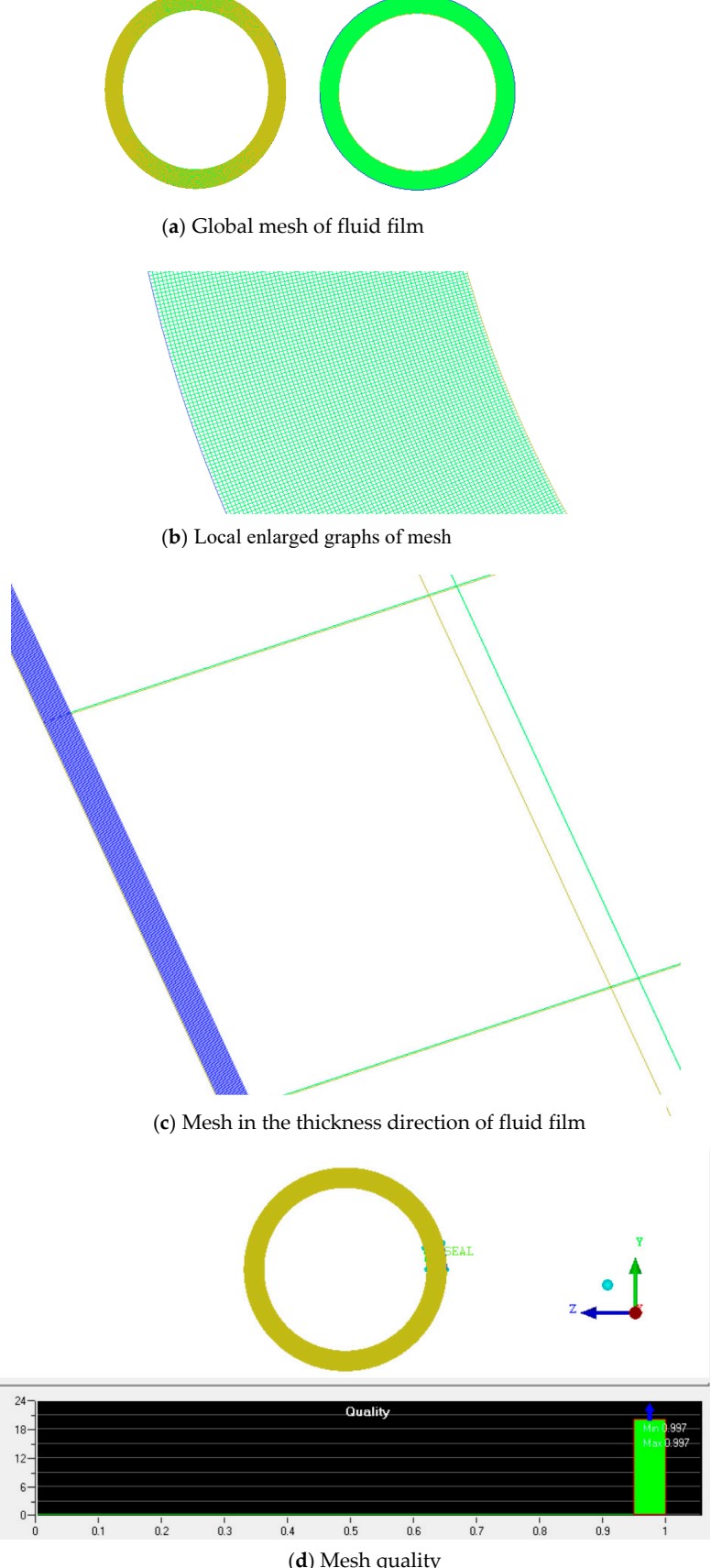

(**a**) Global mesh of fluid film

(**b**) Local enlarged graphs of mesh

(**c**) Mesh in the thickness direction of fluid film

(**d**) Mesh quality

**Figure 11.** Mesh of fluid film and mesh quality.

Mechanical seal material and boundary condition were set as followed:

The rotating ring material used in this paper is silicon carbide, and the stationary ring material is carbon graphite. The main properties of the materials used in the rotating and stationary rings are shown in Table 2.

**Table 2.** The material properties of mechanical seal.

| Properties | Rotating Ring | Stationary Ring |
| --- | --- | --- |
| Material | Silicon Carbide | Carbon Graphite |
| Density $\rho$ (kg/m$^3$) | 3150 | 1810 |
| Specific heat capacity $c$ (J/kg·K) | 710 | 880 |
| Thermal conductivity $k$ (W/m·K) | 150 | 45 |
| Thermal expansion coefficient $\alpha$ (1/K) | $4.3 \times 10^{-6}$ | $6.2 \times 10^{-6}$ |
| Poisson ratio | 0.27 | 0.26 |
| Elastic modulus $E$ (GPa) | 380 | 25 |

The details about boundary, mesh, and calculation are listed in Table 3.

**Table 3.** Details about boundary, mesh and calculation.

| Type | Details |
| --- | --- |
| Entrance boundary | pressure inlet |
| Exit boundary | pressure outlet |
| Wall boundary | standard wall functions |
| Mesh quality | 0.9 |
| Grid number | 362,943 |
| Flow state | laminar flow |
| Algorithm | Simple C |
| Solver | Steady-state solver |

When the residual values of all variables are reduced to $10^{-3}$, the calculation converged.

Entrance boundary conditions in numerical simulation: the pressure inlet boundary condition was adopted, and the inlet boundary position was set outside the fluid film.

Exit boundary conditions in numerical simulation: the pressure outlet boundary condition was adopted, and the outlet boundary position was set inside the fluid film.

Wall boundary conditions in numerical simulation: Standard wall functions were used.

It had been a hot issue that the fluid film flow is laminar flow or turbulent flow in the mechanical seal field. The fluid coefficient $\alpha$ method was adopted to determine whether it is laminar or turbulent [28] in this paper.

$$\alpha = \sqrt{(\frac{\text{Re}_c}{1600})^2 + (\frac{\text{Re}_p}{1600})^2} \tag{9}$$

$\alpha < 0.58$, is laminar flow. $\alpha > 1$, is turbulent flow.

$$\text{Re}_c = \frac{2\pi\rho nrh}{60\mu} \tag{10}$$

$$\text{Re}_p = \frac{\rho Q}{2\pi\mu r} \tag{11}$$

$Q = 1.12 \times 10^{-5}$ kg/s, $n = 1500{\sim}6000$ rpm, $r = 0.031$ m, $h = 1{\sim}9$ μm, $\rho = 998$ kg/m$^3$.

Through calculation could get, $\alpha = 0.29 < 0.58$, the fluid film is laminar flow. Therefore, the laminar model was adopted for the relevant flow dynamic calculation.

The simulation part was set according to the theoretical physical fault model and the dynamic model set up in the Section 3.2.1., and the flow chart of simulation and the analysis details are shown in Figure 12. Simple C algorithm and steady-state solver were used during the calculation of the fluid domain. Firstly, the fluid domain results were calculated by CFD. Then, the fluid domain results were loaded to the solid domain through the Workbench platform. The dynamic equation of mechanical seal axial movement was used during the calculation of the deformation and forces in solid domain. Thus, the solid domain dynamic results, deformation, and forces were obtained. Finally, the fluid film dynamic characteristics, the force, and deformation of the seal rings could be analyzed from the results.

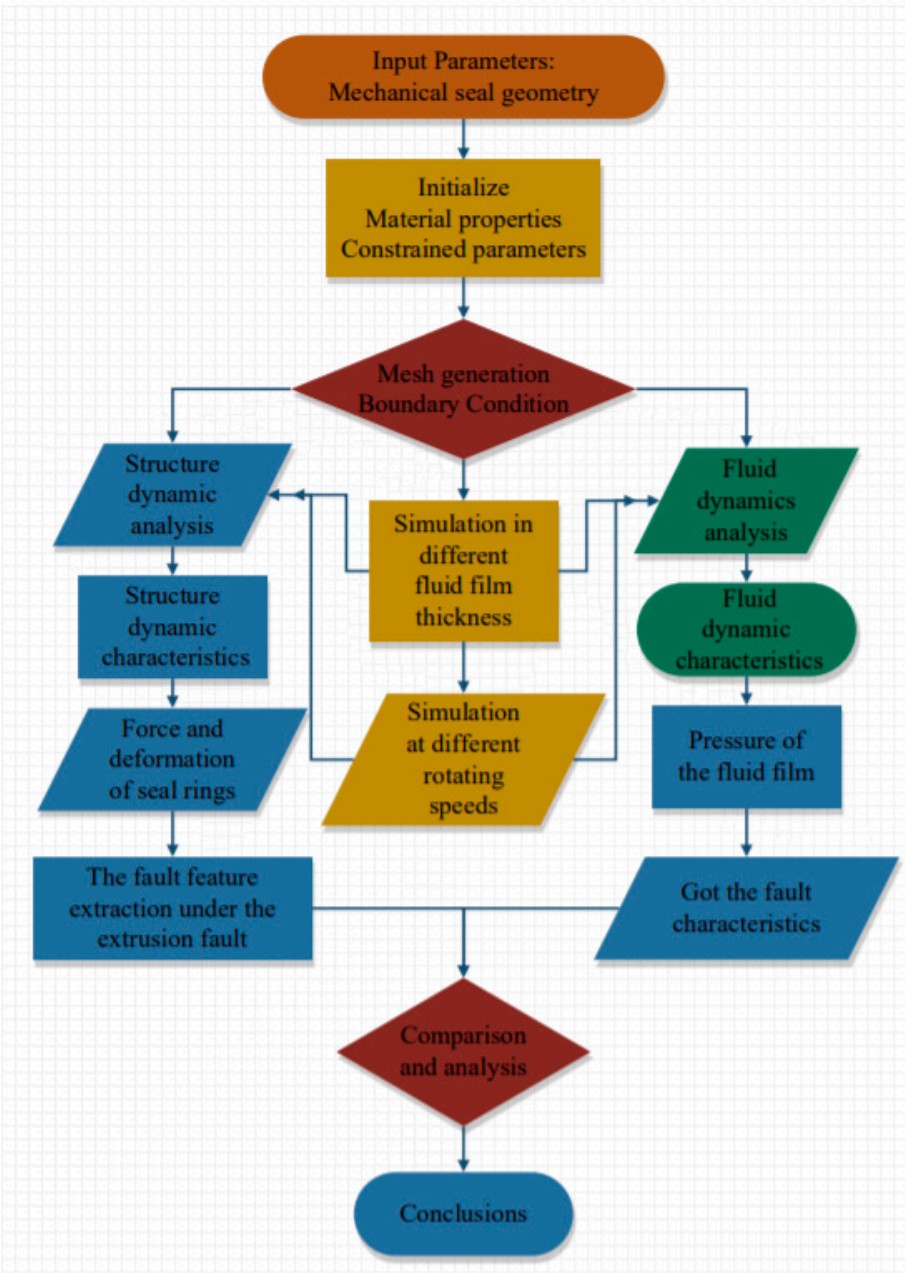

**Figure 12.** Flow chart of calculation.

The first group of simulation process is for different film thickness. The force and deformation simulation data of rotating ring and stationary ring were obtained, and the pressure of fluid film could be got at the same time.

The second group in the simulation process was carried out under different speeds. The hydrodynamic and structural dynamic characteristics were obtained.

Repeat each simulation process. Find the characteristics under different conditions through data analysis.

## 5. Results and Discussion

### 5.1. Fluid Film Pressure Analysis

The pressure distributions under different extrusion degrees are shown in Figure 13. The fluid film pressure would grow as the fluid film thickness increases when the rotating speed $n$ = 2950 r/min, pressure inlet was 0.2 MPa. The fluid film thickness represents the extrusion fault degree. The dynamic characteristic parameters could be reflected by three typical cases. Two-μm fluid film could reflect the serious extrusion fault, 3-μm fluid film was a sign of slight extrusion fault, 4-μm-fluid film was on behalf of the normal state. Such large pressure change between 2-μm fluid film and 3-μm fluid film means that the fluid film thickness has strong recognition capability for the changes of the mechanical seal operation. When the extrusion fault of mechanical seal becomes severer, the fluid film pressure fluctuation would tend to be more intensive. Such tendency indicated that the thickness and the pressure could be treated as the indicator of extrusion fault.

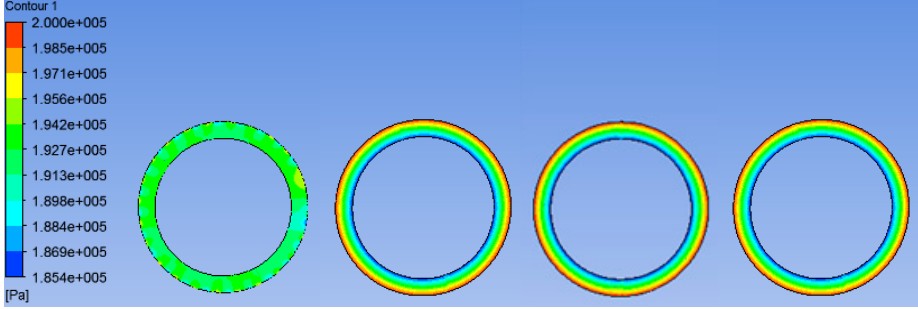

**Figure 13.** The pressure of 2, 3, 4 and 5 μm fluid film.

No significant fluctuation could be found in the 4-μm fluid film pressure contours. This was mainly due to that the 4-μm fluid film was the relatively proper thickness for this mechanical seal. From Figure 13, it could be seen that the pressure was relatively evenly distributed at this time. The thicker fluid film leaded to the bigger viscous shear flow. In addition, pressure bearing capacity could also become stronger with the rapidly increasing thickness. The results indicated that the thicker fluid film could efficiently be used to maintain the hydrodynamic effects and improve the lubrication performance. However, there was deficiency existing in this parameter, which was not sensitive to the light extrusion fault.

The pressure fluctuation under different fluid film is shown in Figure 14. The pressure mainly fluctuates at 0.189 MPa when the thickness of the fluid film is 2 μm. The pressure mainly fluctuates at 0.192 MPa when the thickness of the fluid film is 3 μm. The pressure mainly fluctuates at 0.195 MPa when the thickness of the fluid film is 4 μm. The pressure mainly fluctuates at 0.198 MPa when the thickness of the fluid film is 5 μm. The pressure fluctuation is largely affected by the operation of the rotating ring and stationary ring. When there is extrusion fault, the operation of the rotating ring and stationary ring will change. The pressure distribution and pressure fluctuation of the fluid film will change with the fault. To a certain extent, the pressure fluctuation could reflect the occurrence of extrusion fault. Besides, the fault degree could be preliminarily judged according to the pressure distribution and pressure fluctuation within a certain range.

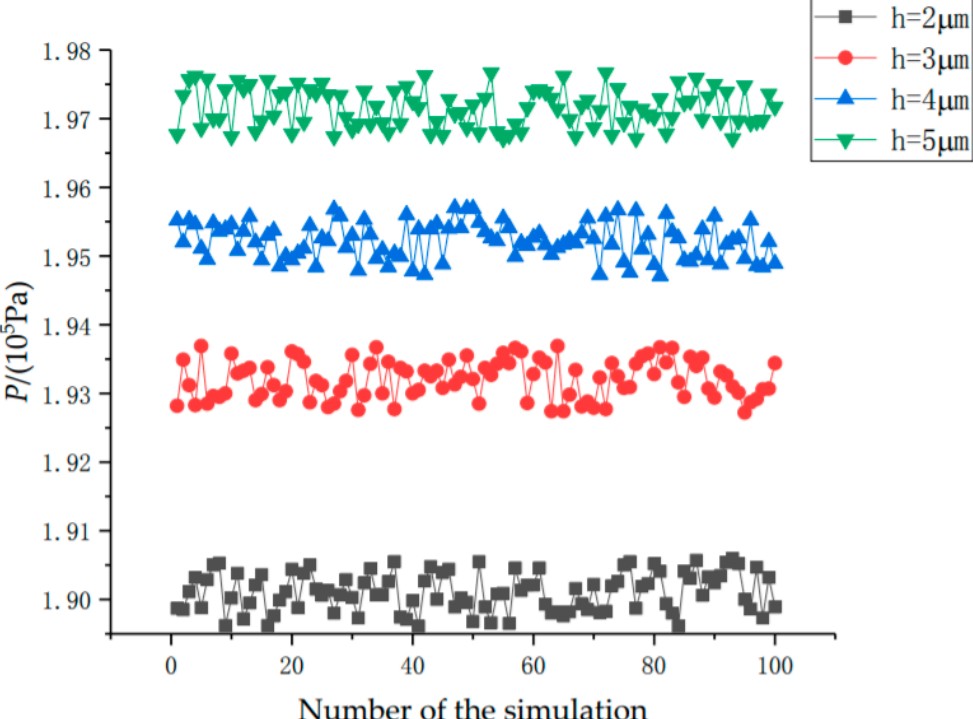

**Figure 14.** Pressure fluctuation diagram under different fluid films.

## 5.2. Seal Performance Analysis

Compared with pressure, leakage $Q$ showed a stronger ability in reflecting the slight extrusion fault. That was mainly due to the leakage as the main indicator that could measure seal fault. Figure 15 showed that the leakage increases with fluctuations when the fluid film becomes thicker. Besides, the minimum value of the leakage, shown in Figure 15, under different fluid film thickness. As shown in the red circle in Figure 15a, too thin fluid film could lead to the severe wear. The leakage curve reflected that the leakage increasing rate is the slowest when the fluid film thickness is between 5 and 6 μm. It depicted that this fluid film thickness was the most beneficial to mechanical seal. So, the best thickness area for this mechanical seal was from 5 to 6 μm. However, the increasing tendency is obvious when the fluid film thickness was more than 6 μm in the red circle in Figure 15b. It could be concluded that faults have occurred on the mechanical seal. With the development of faults, the leakage curve would still increase. According to Figure 15, a large amount of leakage resulted from too thick fluid film, which was directly related with the extrusion fault. During the simulation operation of this mechanical seal, the extrusion fault was more likely to occur when the fluid film thickness was too large or too small. Dynamic characteristics of fluid film would change when the extrusion fault occurs. Hydrodynamic effects were the typical parameters to measure the sealing performance. As the fluid film thickness beyond the best thickness area marked with red circle in Figure 15c, the hydrodynamic effects of fundamental frequency became weaker, while volume force and inertia force got larger.

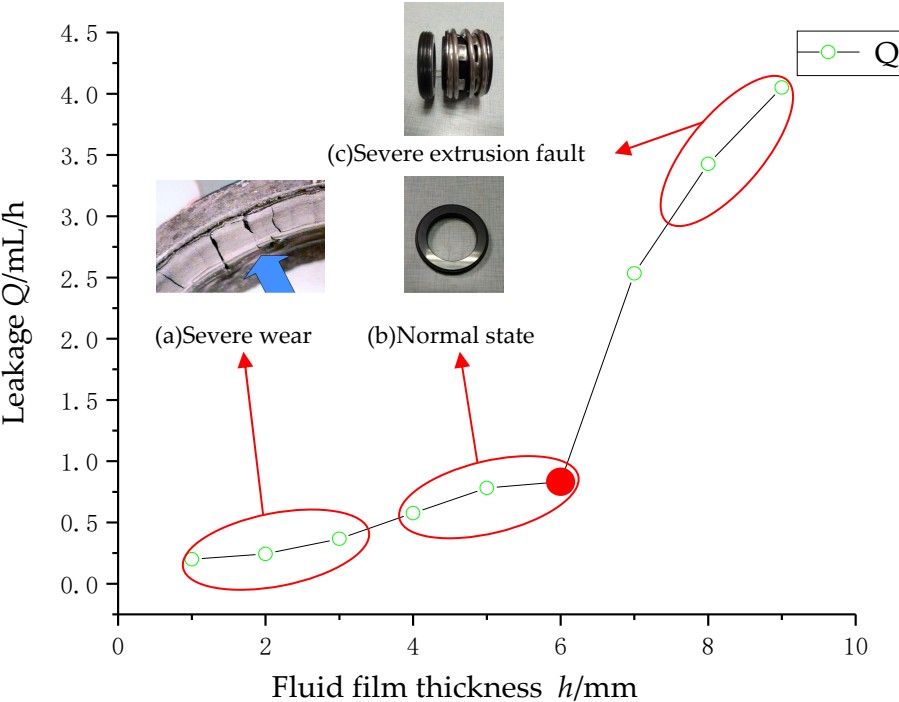

**Figure 15.** Leakage under different fluid film.

The thickness of fluid film was set from 1 to 9 μm. The opening force changed from 60 N to 87 N. The rotating speed was set as 2950 r/min. Figure 16 illustrated that with the increasing of fluid film thickness, opening force became larger rapidly. Moreover, the opening force would have a fluctuating downward trend as the fluid film thickness increases. The opening force could be regarded as the important parameter for the seal dynamic characteristics. As shown in the Figure 16, the opening force was small when the fluid film thickness was less than 3 μm. It was because that serve extrusion fault occurs when the thickness was too thin. The mechanical seal surface friction would happen directly when the thickness was too small. Thus, there would be less or no hydrodynamic effect in this situation. A downward trend of opening force was shown in curve when the fluid film thickness was more than 4 μm. Moreover, the opening force was smaller while the thickness is larger. It indicates that serious damage has occurred on the surface of the rotating ring and stationary ring with the increasing of the thickness. The opening force could have relationship with the fluid film lubrication status and seal extrusion fault degree.

Besides, cavitation could be found in Figure 16b,c. With the increase of fluid film thickness, cavitation phenomenon is further strengthened. It is also due to the cavitation phenomenon that the opening force increases firstly and then decreases with the increase of film thickness. The pressure reduction of mechanical seal is mainly due to the local separation caused by narrow gap, micro groove machined on the surface and surface roughness, and vortex caused by micro modeling. In the conventional scale flow, the surface roughness of micro channel, which is often neglected due to its small influence, has a significant influence in the micro channel flow. The micro disturbance caused by the surface roughness often affects the flow at the edge of the fluid film, which is also one of the main reasons for the cavitation of the micro gap fluid film in the mechanical seal.

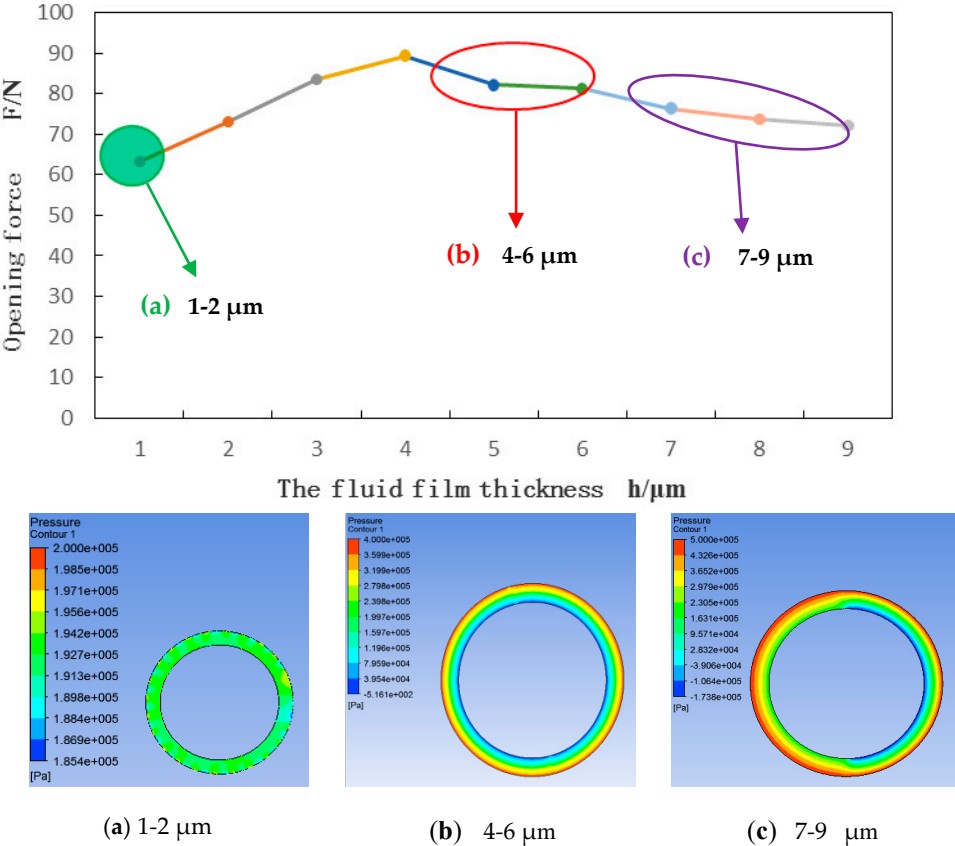

**Figure 16.** Opening force under different fluid film.

As shown in Figure 17, with the increasing of rotating speed, the leakage grew with many fluctuations. The Figure 17a could reflect the same trend of leakage under different film thickness, and Figure 17b could reflect the difference under each speed. It could be seen that the most obvious growth occurs when the rotating speed *n* from 2500 r/min to 3500 r/min. This was mainly due to that the mechanical seal was from normal operation state to the fault status. The inherent factors could be the opening force and friction torque, which resulted in the rapid growth of leakage. Thus, the leakage could be related to the rotating speed. The fluid film stiffness would not always increase with the rotating speed growth, because the growth of the friction torque was suppressed to a certain extent. Based on the combined action of the hydrodynamic and fluid film stiffness, the seal leakage could reflect the sealing performance and the seal faults.

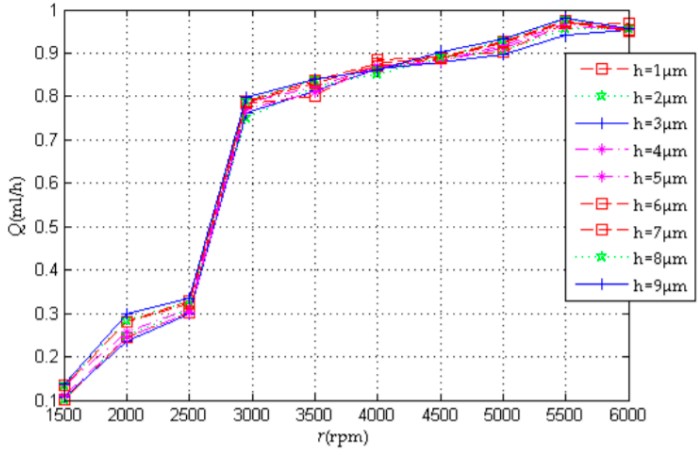

(**a**) Leakage under different film thickness

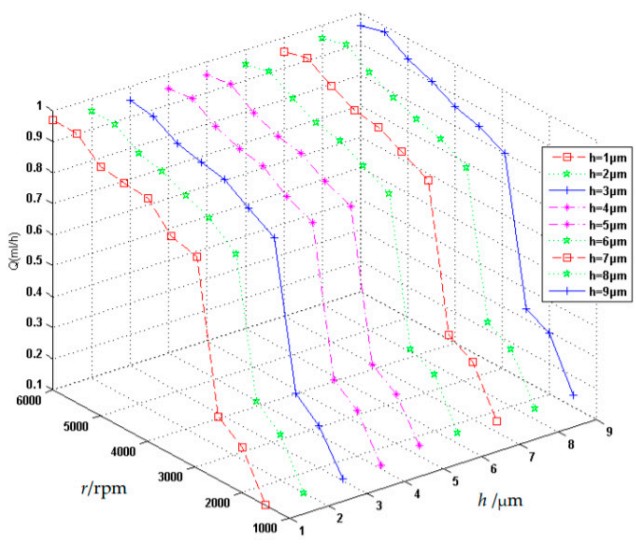

(**b**) Leakage at different speeds

**Figure 17.** Leakage under different rotating speeds and fluid film thickness.

## 6. Conclusions

The dynamic characteristics of mechanical seal under different fault conditions were processed and analyzed in this research. Several conclusions could be drawn from the results described above.

1.  It was the first time to do research on the dynamic characteristics of mechanical seal in the different fault degrees, and proved feasibility of this method. Meanwhile, the leakage *Q*, opening force *F*, rotating speed *n* were combined for the research on seal performance, fault mechanism, and fault degrees analysis.
2.  The leakage analysis of mechanical seal in the different degrees of extrusion fault was conducted with the increasing of fluid film thickness. The extrusion fault is more likely to occur when the fluid film thickness is too large or too small. As the fluid film thickness beyond the best thickness area, the hydrodynamic effects of fluid film turn weaker, while volume force and inertia force get larger. Such tendency has the reference value of mechanical seal fault detection.
3.  The leakage and opening force present obvious change of fluid film thickness. With the increasing of rotating speeds, the leakage grows with many fluctuations. Based on the combined action of

the hydrodynamic and fluid film stiffness, the seal leakage could reflect the sealing performance and the seal fault. There must be a law between the extrusion degrees and fluid film thickness. This paper researched the law that too thin or too thick fluid film would result in the heavy extrusion fault. The fluid film stiffness, leakage, and opening force are the important parameters, which have recognition capability for the extrusion degrees of the mechanical.

This research work has proved the feasibility that the dynamic characteristics of mechanical seal could be found to reflect the degree of extrusion fault. Besides, the research conclusions could have the reference value of mechanical seal fault detection. Further research should focus on the different type of the mechanical seal and find the accurate correspondence relationship.

**Author Contributions:** Conceptualization, Y.L.; methodology and software, Y.L. and Y.F.; validation, Y.L. and Y.F.; formal analysis, Y.L. and Y.F.; investigation, Y.F. and W.Z.; data curation, Y.L. and Y.F.; writing—original draft preparation, Y.L. and Y.F.; writing—review & editing, Y.L., Y.H. and E.A.; project administration, Y.L. All authors have read and agreed to the published version of the manuscript.

**Funding:** This research was funded by the National Natural Science Foundation of China, grant number 51979127, the Nature Science Foundation of Jiangsu Providence, grant number BK20171403, and German KSB Global Headquarters Research Fund, project number: 1.2018.07.1

**Conflicts of Interest:** The authors declare no conflict of interest.

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
