# Peer review of "Research on the Dynamic Characteristics of Mechanical Seal under Different Extrusion Fault Degrees"

_processes, doi:10.3390/pr8091057_

Round 1
Reviewer 1 Report
The reviewer wants to thank the authors for their paper presenting a numerical study investigating different mechanical seals including faults. S/he has some comments/questions/suggestions:
*1) Abstract Keywords: The authors didn’t mention which numerical code they are using. Based on the figures and the mentioning of the Workbench, the reviewer assumes that it will be ANSYS products. Please clarify if this as well as if you use Fluent or CFX for the fluid.
*2) Line (L) 32/33/34: Please either use % or percent constantly.
*3) L 35: German Engineering Association. …. Please add a specific reference.
*4) L 38+40: In two references the authors mention 9 paper. Please split those and explain why each one or at least a smaller group is referenced.
*5) L43: Instead of W.H [14] the authors should use He et al. [14]… please check the complete paper. Again in L48,51 and 55
*6) Figure 2: Is this really a diagram? The reviewer would name this pictures?
*7) Figure 3: Please expand the caption of the figure and also check the other ones. Ideally, the reader should understand it without the text.
*8) Figure 4+5: The drawings on the right side doesn’t fit to the extrusion, at least for Fig.4. Please show those side by side.
*9) Figure 6: not really useful. Please delete or integrate in another one. Especially, the colour scale on the right one is missing.
*10) L188: this sentence is confusing.
*11) L216/220: Please use p or P and if they are different pressure please use an index. Fluid pressure in L216 and pressure of fluid film in L220. Similar issue in L255
*12) Figure 8: The outlines don’t fit the ones in Fig 4+5? Could you please integrate those figures in one figure?
*13) L239/240: Those two sentences are confusing.
*14) Figure 9: without a colour bar those figures are not very useful.
*15) Figure 10 and 12 should be integrated in one.
*16) Figure 11: Both colours are labelled with pressure. Please clarify this and also provide the exact parameters for the mesh test (geometry, speed,...)
*17) Figure 13: The size of the three rings seems to change. Are they manually joint together? Please use the ANSYS-Post and import all three cases and export one figure. Why didn’t the authors show the 5ym option?
*18) Figure 16: Similar issue like before. Is the diameter changing? If yes, please clarify this in the caption?
*19) Figure 16 II: The negative pressure is -1.7*10^5. What about cavitation or did the reviewer missed something?
*20) Figure 17: Should be in 2D and don’t use similar colours and markers.
The reviewer will read the paper again in detail. Thank you.
Reviewer 2 Report
The paper “Research on the Dynamic Characteristics of Mechanical Seal under Different Extrusion Fault Degrees” submitted to the Processes journal is interesting and worth to publish. The content is good prepared and divided into logical sections. I suggest to publish the paper after a minor revision. Specific comments are below.
The mesh of the fluid zone presented in Fig. 6 is not good visible. It would be good to present a cross section of the mesh in a plane perpendicular to the plane of the sheet. The same remark applies to Figure 9.
Was the so-called inflation mesh in the fluid zone used?
Have I good understand that the ANSYS software in the investigations was used? If yes, the Fluent or the CFX code in CFD analysis was used? Which other modules of the ANSYS software were applied?
In the model the energy equation was used. However, in the Results and Discussion section the thermal effects are not commented. Please to add comments related to this issue.
In which way the heat transfer between the fluid and the solid zone was modelled?
There are errors in label descriptions in Figures 11 (Mpa), 14, 15 and 16.
In Fig. 14 the pressure fluctuation in time is presented. How is it possible since the steady-state solver was used?
It is difficult to see differences in the functions shown in Fig. 17. All of them seems to be the qualitatively same. Please to consider another visualisation of these data.
Editorial bugs in descriptions of units may be found in a few places of the paper. The denotation is not standardized. Units are sometimes in parentheses and sometimes not.
A paragraph in conclusions is two times repeated.
Reviewer 3 Report
Review of the manuscript entitled: Research on the Dynamic Characteristics of Mechanical Seal under Different Extrusion Fault Degrees.
The reviewed paper is quite interesting, but needs major corrections and extensions in methodology, presentation of data and explains in a compare numerical and experimental results. Some problems to understand results is missed important information about prepare simulation model and done provide the numerical calculations. Below main lacks and elements for correction:
1. Provide more details about the simulation tool, boundary, and initial conditions of simulation, mesh quality parameters. What is, for example, the value of y+ for this case? The thin-film domain needs good quality mesh.
2. Provide information about the convergence criteria of simulation.
3. This case should be modeled as FSI. Please provide more information about fluid and solid domain interactions.
4. What about compare results? Each simulation data must compare to experimental results. The verification of simulation results needs to be shown in the same physical properties. For example, if you present contour maps of the pressure distribution from CFD, you must compare it to the same from experimental measurement data.
5. The manuscript should be prepared according to the journal template. Check and make corrections of the resolution of figures, line numbering, etc. Figs 6, 9, 13, 16 (mesh views and pressure distribution contours) are completely unreadable.
Round 2
Reviewer 1 Report
The reviewer wants to thank the authors for their answers and corrections. Some points are still open:
*1) Please add Fluent to the abstract as well as the keywords. The reviewer also doesn’t use the newest version of ANSYS but 17 was introduced in 2016. Four years are a long time in the software development. Please consider upgrading.
*2) Please reference what you have. If it is only a report that is fine but make it clear where the information comes.
*3) Figure 6 is not very useful. Please delete it or add very specific information like the boundary conditions….
*4) Equation (2): p and in Eq. (3) P. Is this the same variable, or not.
*5) Figure 9: (a) what is the difference in colour from left to right. Please add the colour bar. (a-d): please export the picture from ANSYS with a white background and not with the standard blueish and switch on the length scale. Labels should be readable.
*6) CAVITATION: This cannot be dealt with just one sentence. Please clarify clearly where cavitation occurred and how the authors dealt with the unrealistic negative pressure (It can be excluded with a step function and a user defined equation but better would be to introduce a cavitation model). All force evaluations are wrong hence ANSYS simply includes those negative values and based on this the results are not realistic. This is a mayor issue and has to be dealt carefully.
*7) Figure 17: Why are not the same line colours and markers used for a and b.
Thank you.
Reviewer 3 Report
Review of the manuscript entitled: Research on the Dynamic Characteristics of Mechanical Seal under Different Extrusion Fault Degrees
Thank you for your response. In some points, I agree. The reviewed paper has provided information that will more useful for readers.
The authors did simulations, analyzing the results, and preparing the manuscript. Therefore, it would be worthwhile to extend the reference recognition by several publications on similar cases of modeling thin-film flow. It will a good way to show the relevance of prepared simulations. The subject should be noticed if will be supported by extensive recognition of references from the last 10 years. For example please check and try to follow the paper DOI: 10.15199/62.2015.12.3. It's an example of the analysis of thin-film flow and share stress analysis. Of course, you can find many other examples. Try to add some more to expand the introduction and discussion sections.
